# PLATE-Seq for genome-wide regulatory network analysis of high-throughput screens

Erin C. Bush[1,2], Forest Ray[1], Mariano J. Alvarez[1,3], Ronald Realubit [1,2], Hai Li[1,2], Charles Karan[1,2], Andrea Califano[1,2,4,5] & Peter A. Sims [1,2,4]

Pharmacological and functional genomic screens play an essential role in the discovery and characterization of therapeutic targets and associated pharmacological inhibitors. Although these screens affect thousands of gene products, the typical readout is based on low complexity rather than genome-wide assays. To address this limitation, we introduce pooled library amplification for transcriptome expression (PLATE-Seq), a low-cost, genome-wide mRNA profiling methodology specifically designed to complement high-throughput screening assays. Introduction of sample-specific barcodes during reverse transcription supports pooled library construction and low-depth sequencing that is 10- to 20-fold less expensive than conventional RNA-Seq. The use of network-based algorithms to infer protein activity from PLATE-Seq data results in comparable reproducibility to 30 M read sequencing. Indeed, PLATE-Seq reproducibility compares favorably to other large-scale perturbational profiling studies such as the connectivity map and library of integrated network-based cellular signatures.

---

[1] Department of Systems Biology, Columbia University Medical Center, New York, NY 10032, USA. [2] Sulzberger Columbia Genome Center, Columbia University Medical Center, New York, NY 10032, USA. [3] DarwinHealth Inc., New York, NY 10032, USA. [4] Department of Biochemistry & Molecular Biophysics, Columbia University Medical Center, New York, NY 10032, USA. [5] Institute for Cancer Genetics, Herbert Irving Comprehensive Cancer Center, Columbia University Medical Center, New York, NY 10032, USA. Erin C. Bush and Forest Ray contributed equally to this work. Correspondence and requests for materials should be addressed to A.C. (email: andrea.califano@columbia.edu) or to P.A.S. (email: pas2182@cumc.columbia.edu)

High-throughput screening (HTS) represents a key component of drug discovery and a critical technology used throughout biomedical research[1, 2]. Due to cost and complexity, however, most screens are still performed using low-complexity reporters, such as cell viability, inhibition of specific enzymes, binding affinity, protein-protein interactions, mitochondrial respiration, and cell growth. In contrast, there is increasing interest in coupling HTS with genome-wide reporters, which may provide a more comprehensive portrait of drug activity[2–4].

Indeed, a key advantage of genome-wide reporters is their more universal nature; specifically, the ability to test hypotheses that may not have been considered at the time of assay development. For instance, the same data from gene expression profiling of cellular perturbations have been used effectively to predict compound mechanism-of-action (MoA)[5–7] and sensitivity in specific cellular contexts[8], as well as to identify synergistic drug combinations[9, 10], compounds with similar MoA[11], and candidates for drug repositioning[7]. Indeed, when incorporated in an HTS setting, genome-wide profiles can report on virtually any genes or pathways of interest, without requiring an a priori commitment.

Unfortunately, due to their relatively high cost and labor-intensive nature, genome-wide expression profiles have not been incorporated as primary reporters in HTS campaigns. A few exceptions, such as connectivity map (CMap)[2], represent proof-of-concept studies rather than scalable approaches and have either been restricted to a handful of cell lines or replaced by methodologies that report on a limited number of genes (e.g., Luminex L1000 reporters)[3].

To address this challenge, we introduce a new approach that combines a highly scalable and multiplexed RNA-Seq protocol (PLATE-Seq) with regulatory network analysis. Collectively, this integrative and fully automated pipeline allows accurate, reproducible characterization of the proteins, whose activity is affected by a library of bioactive compounds. The proposed approach involves two key concepts: (a) a strategy for barcoding and pooling cDNA libraries to substantially reduce the cost and complexity of multi-sample RNA-Seq and (b) the use of network-based algorithms for the highly reproducible inference of protein activity from low-depth RNA-Seq profiles (0.5–2 M reads). Taken together, this combination supports a >tenfold cost reduction with virtually no reduction in assay accuracy and reproducibility, compared to standard depth (30 M read) sequencing. This translates into a dramatic increase in gene reporter dimensionality for HTS applications—from a few observables to a genome-wide repertoire—at a total reagent cost of ~$15 per sample.

## Results

**PLATE-Seq technology.** Recent advances in multiplexed and single-cell RNA-Seq have led to significant increases in cost effectiveness and scalability of gene expression profiling[12–17]. These methodologies introduce sample-specific sequence barcodes into cDNA prior to library construction, allowing early pooling of cDNA from many samples and a proportional decrease in reagent and labor costs. They are also optimally suited to automation, allowing straightforward integration into HTS pipelines for analysis of RNAi or small molecule perturbations in a multi-well format. Here we use automated liquid handling to

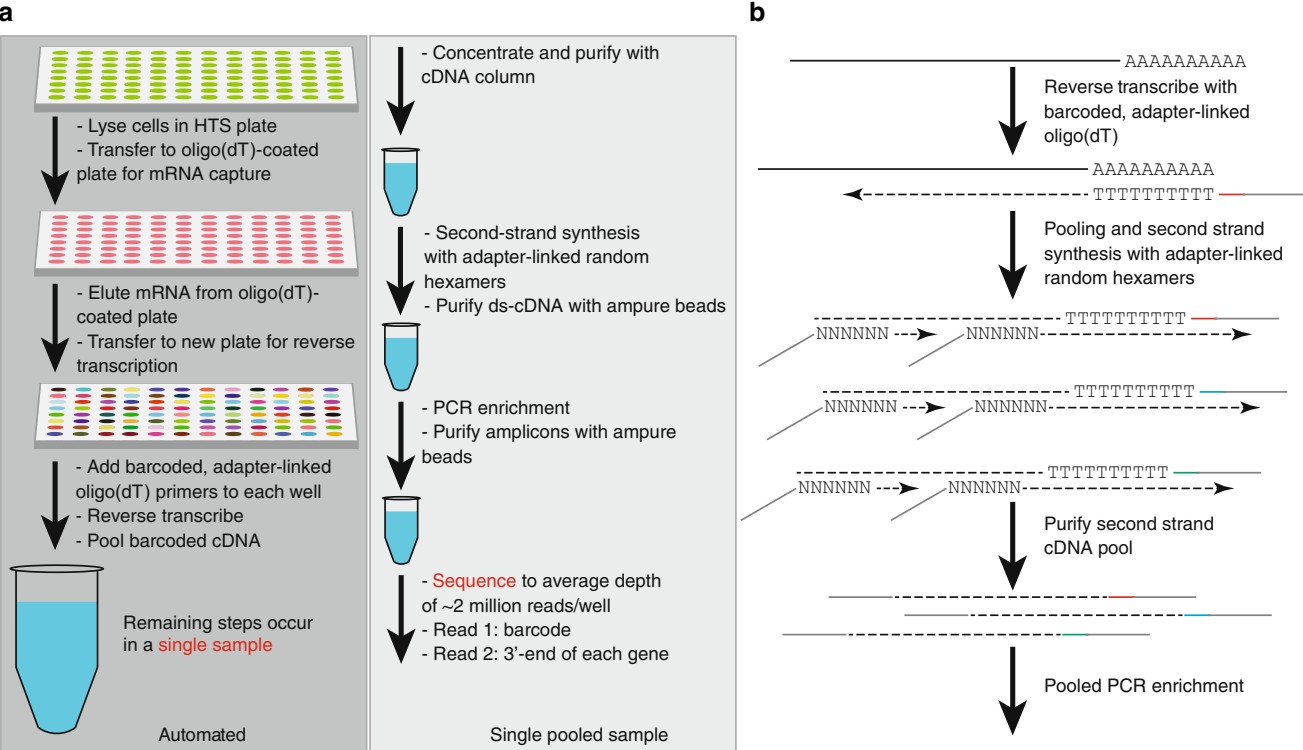

**Fig. 1** Schematic illustration of PLATE-Seq workflow. **a** After conducting a screen in multi-well plates, we lyse the cells and capture mRNA from the cell lysate using an oligo(dT)-coated capture plate. The purified mRNA is then reverse transcribed with barcoded, adapter-linked olig(dT) primers and the resulting cDNA is pooled. All of these steps are automated. The remaining steps, which take place on a single pooled sample, are conducted manually and include cDNA purification, second-strand synthesis, and PCR enrichment. **b** Molecular-level schematic for constructing 3′-end PLATE-Seq libraries. After reverse transcription with oligo(dT), second-strand synthesis of the pooled cDNA is accomplished using random hexamer primers prior to PCR enrichment of the barcoded pool

introduce lysis buffer, capture polyadenylated mRNA with an oligo(dT)-grafted plate, and deliver well-specific, barcoded oligo (dT) primers to every sample in a multi-well plate (Fig. 1a). After reverse transcription, the cDNA in each well contains a specific barcode sequence on its 5′ end and a common adapter, such that all samples can be combined into a single pool for purification and concentration. We then use Klenow large fragment for pooled second-strand synthesis from adapter-linked random primers. Because this polymerase lacks strand displacement and 5′ to 3′ exonuclease activities, each cDNA molecule produces at most, one second-strand synthesis product containing the sample barcode (Fig. 1b). Finally, the pooled library is enriched in a single

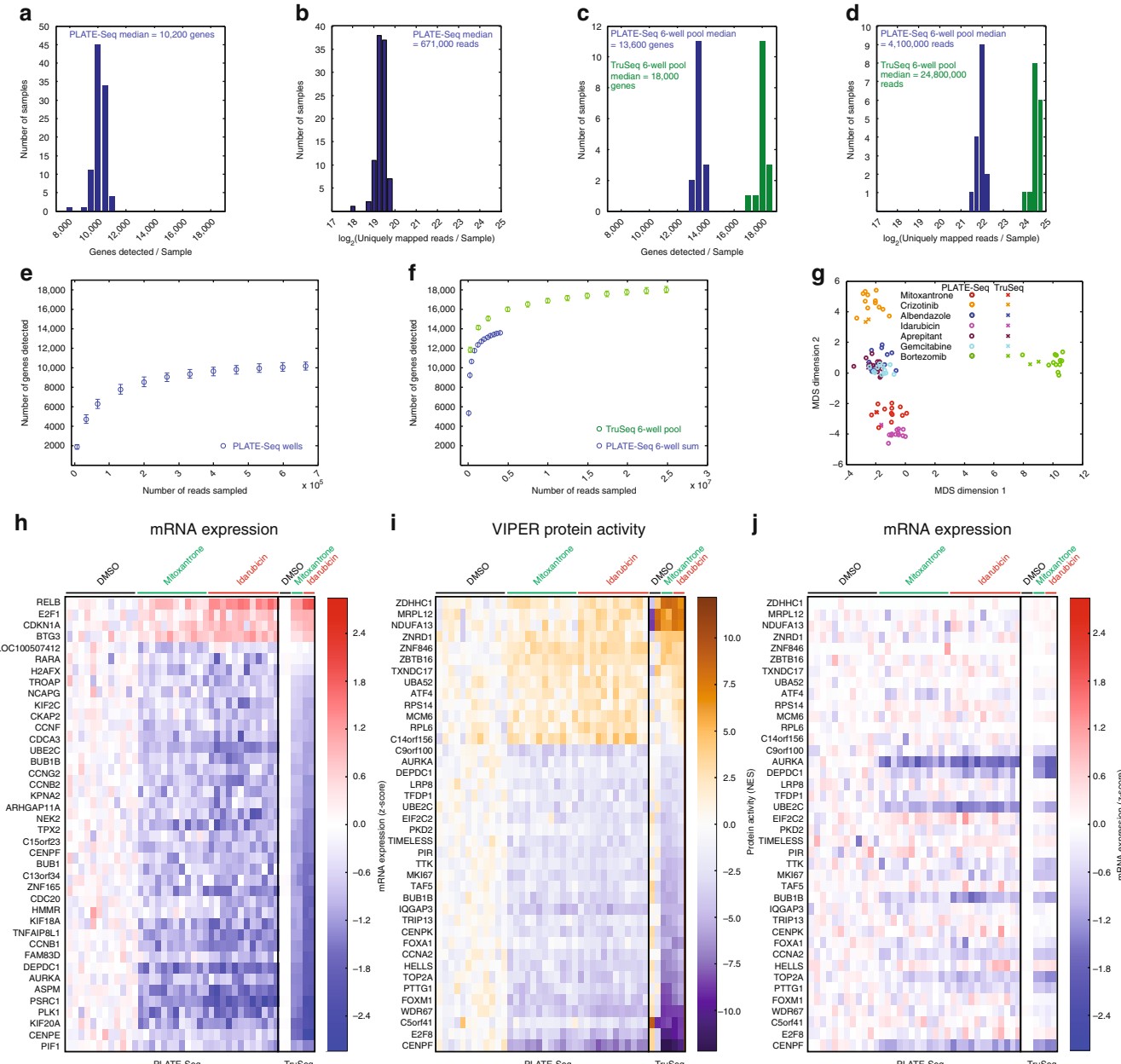

**Fig. 2** PLATE-Seq performance. **a** *Histogram* of genes symbols detected per sample for a 96-well PLATE-Seq experiment in BT20 cells. **b** *Histogram* of uniquely mapped reads per sample for the experiment in **a**. **c** We pooled half of the sample from every six wells for conventional RNA-Seq with 30 M raw reads (Illumina TruSeq). Here we show a *histogram* of gene symbols detected per sample for each six-well TruSeq pool and for the sum of the corresponding six PLATE-Seq samples. **d** Same as **c** for uniquely mapped reads per sample. **e** Gene detection saturation curve for PLATE-Seq samples based on random subsampling. The points represent the average over all 96 wells and the *error bars* are deviations s.e.m. **f** Same as **e** but for each six-well TruSeq pool and for the sum of the corresponding six PLATE-Seq samples. **g** MDS clustering of PLATE-Seq and TruSeq samples based on differentially expressed genes identifying using the PLATE-Seq replicates for each drug compared to vehicle control samples. The PLATE-Seq replicates for each drug cluster together and also with the corresponding TruSeq samples. **h** *Heat map* showing the top 40 most differentially expressed genes based on PLATE-Seq of mitoxantrone- and idarubicin-treated BT20 cells measured with both PLATE-Seq and TruSeq. The two drugs are both topoisomerase II inhibitors and have similar gene expression signatures. **i** Same as **h** but with differentially active proteins as inferred using VIPER. Note that TOP2A, the gene that encodes the target of the two drugs, is strongly deactivated. **j** Gene expression of differentially active proteins inferred using VIPER. Most of these genes are not differentially expressed and some are difficult to detect with PLATE-Seq, yet VIPER can still infer their activities

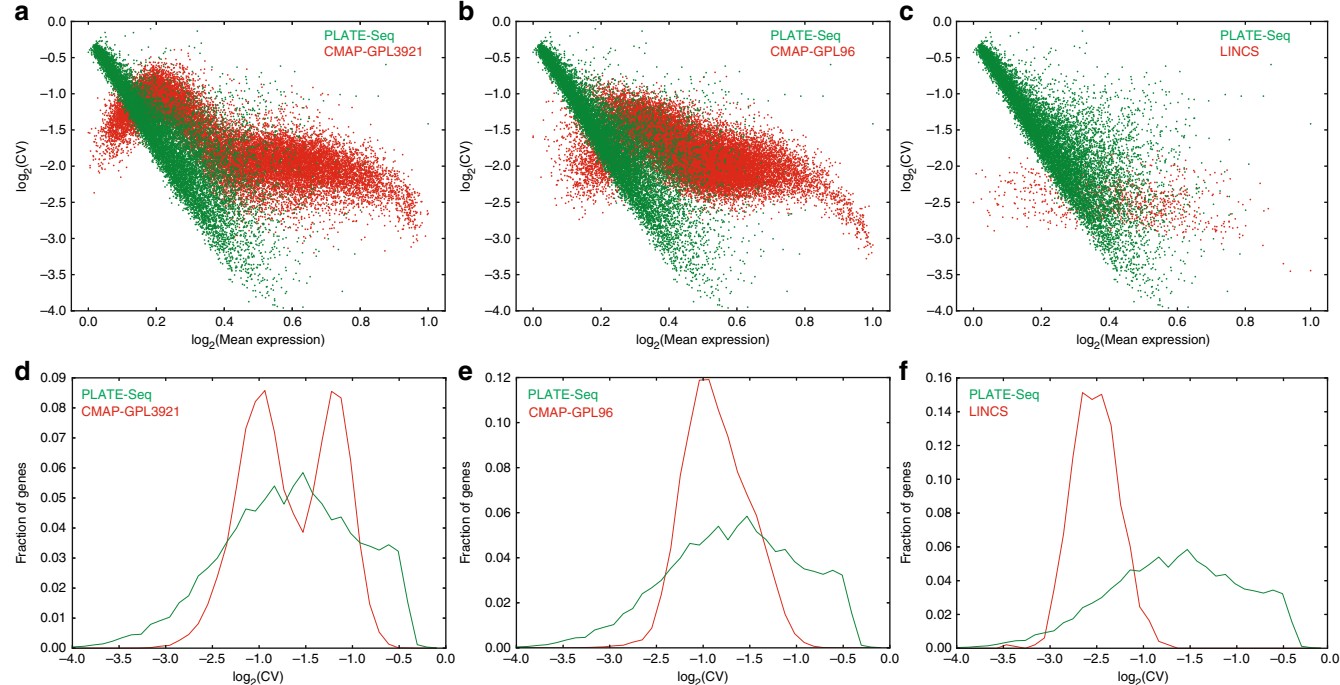

**Fig. 3** Comparison of PLATE-Seq to CMap and LINCS. **a** Mean CV vs. relative mean expression for PLATE-Seq U87 184-compound screen and CMap GPL3921 microarray platform duplicates. **b** Same as **a** for PLATE-Seq U87 screen and CMap GPL96 microarray platform. **c** Same as **a** for PLATE-Seq U87 screen and LINCS/L1000 screen. **d** CV *histogram* across genes for duplicates in the PLATE-Seq U87 screen and CMap GPL3921 microarray platform. **e** Same as **d** for PLATE-Seq U87 screen and CMap GPL96 microarray platform. **f** Same as **d** for PLATE-Seq U87 screen and LINCS/L1000 screen

PCR prior to sequencing. The resulting libraries represent the 3′ ends of mRNAs and are sequenced to a depth of 0.5–2 M raw reads per sample.

**PLATE-Seq performance**. To characterize the performance of PLATE-Seq, we conducted a fully automated, 96-well screen to profile BT20 breast cancer cells following treatment with seven well-characterized small-molecule perturbagens (plus DMSO controls) and 12 replicates per condition. Figure 2a, b shows the detected gene and uniquely mapped read distributions across the 96-well plate samples, respectively. On average, we detected ~10,200 genes per sample from ~670,000 uniquely mapped reads from ~10,000 cells per well. To compare PLATE-Seq to conventional RNA-Seq, we processed some of the replicates using the Illumina TruSeq protocol, to a standard depth of ~30 M reads per sample. Critically, TruSeq required pooling six samples to meet the minimum cDNA input requirements.

Figure 2c, d shows the detected gene and uniquely mapped read distributions for both TruSeq and PLATE-Seq samples, normalized to the same number of wells by pooling data from the six PLATE-Seq replicates. Figure 2e, f shows the saturation behaviors of individual PLATE-Seq wells, TruSeq samples, and aggregate PLATE-Seq data pools equivalent to the TruSeq replicate pools. Analysis of data pooled from six single-well PLATE-Seq profiles detected >75% of the genes that had been detected in the same number of replicates by TruSeq, albeit with sixfold fewer reads. We also compared the relative efficiencies of conventional column isolation and oligo(dT) bead-based purification of mRNA and the oligo(dT) plate-based purification used here. Not only does the plate-based purification offer significant advantages in terms of automation, the gene detection efficiencies and saturation behaviors of the resulting libraries from the two methods are essentially identical (Supplementary Fig. 3).

We also investigated whether the gene expression signatures associated with each drug were comparable between the two methods. We used DESeq2 (ref. [18]) to identify differentially expressed genes ($q < 0.05$) between each set of drug-treated and vehicle-control replicates measured by PLATE-Seq and projected a matrix of fold changes computed from both PLATE-Seq and TruSeq across these genes using multidimensional scaling (Fig. 2g). As expected, replicates associated with distinct drug treatments cluster together, consistent with their tissue-specific activity and independent of profiling technology. For instance, drugs such as aprepitant, gemcitabine, and albendazole, which had minimal effects on BT20 cells, were clustered together. In contrast, strongly bioactive drugs, such as crizotinib and bortezomib, produced distinct but highly reproducible and technology-independent clusters. Drugs with similar MoA, such as the topoisomerase II inhibitors mitoxantrone and idarubicin, clustered together. Finally, we performed gene set enrichment analysis (GSEA) of significantly altered genes detected by PLATE-Seq in the TruSeq differential expression analysis. For all gene sets, we found highly significant enrichment of the PLATE-Seq genes in the appropriate direction (Supplementary Fig. 1). Hence, both PLATE-Seq and conventional RNA-Seq produce comparable gene expression signatures following drug perturbation.

In Alvarez et al.[19], we showed that assessing protein activity from gene expression profiles using VIPER is highly robust to low-depth sequencing. Indeed, unlike gene expression profiles, whose correlation degraded rapidly with decreasing sequencing depth, Spearman's correlation of VIPER-inferred activity profiles was virtually unaffected between 0.1 and 30 M reads[19]. This is because VIPER infers protein activity from expression of its tissue-specific transcriptional targets, as identified by the ARACNe algorithm. Thus, VIPER integrates dozens to hundreds of gene expression measurements to infer activity of a single protein, effectively averaging out noise resulting from low-depth sequencing[19].

**Protein activity inference with PLATE-Seq.** To demonstrate the practical utility of this theoretical advantage when using low-depth PLATE-Seq assays, we used VIPER to compute differential activity of ~6000 regulatory proteins, including ~2000 transcription factors and ~4000 signaling proteins, following treatment with the topoisomerase II inhibitors mitoxantrone and idarubicin[19]. Figure 2h shows the gene expression signatures of the top 40 genes most differentially expressed following perturbation by either drug, based on PLATE-Seq. As expected from Fig. 2g, there is excellent agreement between PLATE-Seq and TruSeq. To identify key regulatory proteins responsible for the pharmacological effects of these two drugs and thus comprising their MoAs, we analyzed both the PLATE-Seq and the TruSeq profiles with VIPER. We used our recently published breast cancer-specific gene regulatory network[19] generated by ARACNe analysis[20, 21] of gene expression profiles from the breast cancer cohort collected by The Cancer Genome Atlas consortium[22]. Figure 2i shows the top 40 most differentially activated proteins following mitoxantrone or idarubicin treatment, based on PLATE-Seq. Critically, VIPER-inferred protein activity was highly reproducible across PLATE-Seq and TruSeq samples (see Supplementary Fig. 2 for GSEA). Furthermore, TOP2A, the gene that encodes the target of both drugs, was among the most deactivated regulatory proteins by VIPER analysis, along with several other key cell cycle control proteins (e.g., CEPNF, FOXM1, CCNA2, BUB1B, and CENPK). Finally, Fig. 2j shows the gene expression signatures associated with these top 40 differentially active proteins. Consistent with previous studies[19, 23, 24], many of these proteins, including cell cycle regulators, that play an important role in the MoA of these drugs, such as FOXM1, CCNA2, and TOP2A, are not among the top 40 most differentially expressed genes. This effectively demonstrates that the combination of PLATE-Seq and VIPER analysis produces differential protein activity profiles that are remarkably robust, independent of the low-sequencing depth.

**Comparison of PLATE-Seq to CMap and LINCS.** Previous efforts have attempted to elucidate MoA based on gene expression alterations in cell lines[2, 7]. CMap represents a first seminal attempt at this approach based on microarray profiling of cell lines treated with several hundred compounds[2]. To increase scalability, library of integrated network-based cellular signatures (LINCS) implemented the Luminex L1000, which reports on ~1000 representative markers from which broader expression profiles could be computationally inferred[3, 25]. However, these two data sets, which represent the current state-of-the-art, require a trade-off between cost/efficiency and coverage. Therefore, we reasoned that PLATE-Seq might provide a solution to this trade-off by supporting a genome-wide reporter assay while leveraging the broader dynamic range of RNA-Seq compared to microarrays and the ability to profile >tenfold more genes than the L1000 at a comparable cost.

To compare PLATE-Seq to these technologies, we conducted a 4 × 96-well screen with 184 clinically relevant compounds plus vehicle controls (Supplementary Table 1) in U87 cells in duplicate. We then systematically compared the gene-level noise distribution we obtained to those of the CMap and LINCS/L1000 data sets. Figure 3a–c shows how the average coefficients of variation across replicates vary with average expression level for PLATE-Seq in comparison to CMap data obtained with the GPL3921 microarray platform, CMap with the GPL96 platform, and LINCS/L1000, respectively. Similarly, Fig. 3d–f shows the overall distributions of average coefficients of variation for these same comparisons. Despite the significantly lower costs associated with PLATE-Seq, we obtain comparable noise distributions

to CMap. In addition, we point out that although the noise in the LINCS data set is twofold lower than in the PLATE-Seq screen (for duplicates taken from GSE70138), measurements are only made for 978 genes (>tenfold fewer than in PLATE-Seq, which is a genome-wide measurement). We also show that the dynamic range for signal in PLATE-Seq is comparable to these alternative methods by calculating the fold-change distribution across each expression measurement and perturbation relative to vehicle controls (Supplementary Fig. 4). Hence, PLATE-Seq compares favorably to previous approaches and may offer an effective, low-cost platform for assessing compound MoA similarity.

## Discussion

Small molecule perturbations induce complex, cell-context-specific alterations that ultimately affect thousands of genes. This challenges the use of low-complexity reporter assays in HTS to elucidate MoA. In contrast, RNA-Seq represents a high-dimensional assay that is particularly well suited for compound MoA elucidation[19, 26], similarity analysis[2, 7], and synergy analysis[9] especially when using network-based methods that may pinpoint the key regulatory targets and effectors. However, conventional RNA-Seq involves independent library construction for each sample and deep sequencing, making its application to HTS prohibitively expensive and poorly suited to automation. For instance, at an average cost of $300 per profile, screening a library of ~4000 compounds across two time points, concentrations, and replicates would cost ~$10 M in reagents. However, when combined with automated, pooled library preparation and the tenfold reduction in sequencing depth made possible by leveraging VIPER, the reagent cost for the same 4000-compound campaign would be less than $0.5 M, a 20-fold reduction.

A possible limitation of the approach is its dependence on regulon quality for accurate inference of protein activity. Using large-scale benchmarks, we have shown that differential activity is accurately predicted for >70% of transcriptional regulators and ~60% of signaling proteins[19]. However, if specific proteins of interest are poorly characterized in the assay, ad hoc regulons can be generated by combining DNA-binding assays and expression profiles following perturbation[27]. In addition, as computational reverse engineering methods improve, accuracy of protein activity inference will increase. In this case, data from previous screens can simply be re-analyzed using the improved models to produce more accurate and extensive protein-activity profile coverage, thus avoiding obsolescence of PLATE-Seq data sets.

We expect PLATE-Seq to have broad pharmacological applications, including elucidation of compound MoA and synergy, support for drug re-positioning, and precision medicine approaches where MoA is matched to patient-specific disease dependencies. Finally, while we focused mainly on drug screening, PLATE-Seq is equally amenable to RNAi, CRISPR-Cas9, and cDNA screening assays. This highly scalable and economical approach integrates HTS with high-dimensional analysis of RNA expression and protein activity, revealing the connections between cellular perturbations and genome-wide regulatory interactions.

## Methods

**Cell culture and drug perturbation assays.** BT20 epithelial breast carcinoma cells were cultured in white 96-well tissue culture-treated plates (Greiner 655083) at a starting density of 8000 cells per well in 100 μl of eagle's minimum essential medium supplemented with 10% fetal bovine serum (FBS) and 1% penicillin/streptomycin. After 24 h of incubation, the plates were treated with drugs. Each drug was dosed at the concentration at which the cells were 80% viable after 48 h of treatment. After 6 h of treatment, the medium was replaced with 100 ml of FBS supplemented with 10% DMSO and the plates were frozen at −80 °C prior to PLATE-Seq. After thawing for PLATE-Seq, cells were washed twice with phosphate-buffered saline (PBS) prior to lysis.

**Automated PLATE-Seq**. Cells are lysed for PLATE-Seq in Buffer TCL (Qiagen) supplemented with 2-mercaptoethanol in 96-well plates. We transfer 30 μl of cell lysate from each well to a 96-well plate with oligo(dT) grafted to the wells (mRNA TurboCapture Plate, Qiagen) and added 1 μl of 1:100 dilution of ERCC Ex-Fold Spike-Ins (ThermoFisher) to a subset of the sample wells. The plate is then centrifuged at $295 \times g$ for 1 min to eliminate air bubbles followed by incubation for 90 min at room temperature with shaking at 200 rpm. Lystate is then removed from the oligo(dT) capture plate and wells are washed three times with Buffer TCW (Qiagen) using the 96 CO-RE Head of a Hamilton Microlab STAR automated liquid handling system. After drying the plate, mRNA is then eluted by adding 30 μl of elution buffer (nuclease-free water supplemented with 0.4 U μl⁻¹ SUPERaseIN (ThermoFisher), incubation at 65 °C for 5 min, and incubation at room temperature for 5 min. mRNA samples are then transferred to a new PCR plate (Hard-Shell PCR plate, 96-well, Bio-Rad) using sterile-filter 96-well, 50 μl tips on the Hamilton STAR.

To anneal primers for reverse transcription, 3 μl of 100 μM adapter-linked oligo (dT) primers are added to each well along with 10 μl of 5× ProtoScript RT Buffer (New England BioLabs) with the Hamilton STAR CO-RE Head and 1000 μl channel pipetting system (see Supplementary Table 2 for "PLATEseq_oligodTBC" sequences). The plate is then heated to 94 °C for 2 min and placed immediately on ice for 5 min. The remaining components of the reverse transcription master mix including 0.5 mM of each dNTP, 10 mM DTT, 35.8 U of Protoscript II Reverse Transcriptase (New England BioLabs), and water are then added to bring the final reaction volume to 50 μl per well. The plate is then centrifuged at $295 \times g$ for 1 min and incubated at 42 °C for 2 h. At this point, the samples can optionally be frozen at −80 °C.

To remove excess primer, 1 μl of a fourfold dilution of Exonuclease I (New England BioLabs) is added to each sample. The Exonuclease I reaction is incubated at 25 °C for 1 h. To hydrolyze the RNA, we add 20 μl of a 1:1 mixture of 1 M sodium hydroxide and 0.5 M EDTA to each well and incubate at 65 °C for 15 min. The wells are then pooled together into a single tube to which 80 μl of 12 M hydrochloric acid are added to neutralize the sample. At this point, the pooled sample can optionally be frozen at −80 °C.

The pooled, neutralized sample is purified and concentrated using a Qiagen MinElute column using a vacuum apparatus according to the manufacturer's instructions, and pooled cDNA is eluted in 15 μl of nuclease-free water. For second-strand synthesis, 1 μl of 10 mM dNTP mixture and 1 μl of 100 mM adapter-linked random hexamer primers are added to the pooled cDNA sample (see Supplementary Table 2 for "PLATEseq_second_strand" primer sequence). The mixture is then heated to 70 °C for 2 min and immediately placed on ice for 5 min. Next, 2 μl of NEB Buffer 2 (New England BioLabs) and 1 μl of Klenow large fragment DNA polymerase (New England BioLabs) are added and the reaction is incubated at room temperature for 30 min. The reaction is then stopped by addition of EDTA to a final concentration of 50 μM the double-stranded cDNA is purified with two rounds of bead clean-up using AMPure XP beads (Beckman Counter) at a 1:1 bead-to-sample ratio. Finally, the double-stranded cDNA pool is amplified by PCR using Phusion DNA Polymerase (New England BioLabs) with 0.5 μM Illumina RP1 primer and Illumina RPIx primer (where x is a number indicating the Illumina index). The thermocycling protocol begins with 98 °C for 30 s followed by 10–12 cycles of 98 °C for 15 s, 62 °C for 15 s, and 72 °C for 60 s followed by a 7-min incubation at 72 °C.

Samples are sequenced on an Illumina NextSeq 500 sequencer using 75 cycle version 2 sequencing kits. Custom sequencing primers were used at a final concentration of 300 nM. Read 1 comprises 26 cycles and read 2 comprises either 60 or 66 cycles depending upon whether or not an index read is included. In each experiment, 20–30% PhiX library is added to the sample, which is clustered at a final concentration of 1.6 pM.

**Comparison of PLATE-Seq to conventional RNA-Seq**. To compare PLATE-Seq and conventional RNA-Seq, we conducted a small-scale screen in a single 96-well plate. We treated each well containing cultured BT20 cells with one of seven compounds or DMSO (vehicle), including mitoxantrone, crizotinib, albendazole, idarubicin, aprepitant, gemcitabine, and bortezomib, with 12 replicate wells for each condition. Half of the material in each well was allocated to PLATE-Seq, and 96 libraries were constructed using the automated, pooled procedure described above. For the other half, we pooled two sets of six wells for each condition to generate 16 samples in total (two replicates for each condition). We then extracted total RNA from each pool using an RNeasy column (Qiagen) and subjected the total RNA to Illumina TruSeq poly(A) + RNA-Seq library construction according to the manufacturer's instructions. The resulting 16 RNA-Seq libraries were then sequenced to a depth of ~30 M raw, 100-base single-end reads on an Illumina HiSeq 2500 sequencer.

**PLATE-Seq performance with conventional mRNA isolation**. LnCAP human prostate adenocarcinoma cells were plated on a 96-well tissue culture treated plate (Greiner #655083) at a starting density of 10,000 cells per well in 100 μl of medium (RPMI1640 supplemented with 10% FBS). After 24 h of incubation, the medium was removed and cells were lysed in 30 μl of Buffer TCL (Qiagen) with 1% β-mercaptoethanol. A measure of 15 μl of this lysate was transferred to a new plate and used for automated PLATE-Seq; the remaining 15 μl of lysate in each well was

mixed with 135 μl of Buffer RLT (Qiagen). A measure of 150 μl of 70% ethanol was added to each well in this plate and all 300 μl was transferred onto the RNeasy 96 filter plate (Qiagen) for total RNA extraction according to the manufacturer's instructions, including DNase digestion. The first elution from the columns was in 50 μl of RNase-free water, and the second elution was in 45 μl of RNase-free water.

RNA Purification Beads (Illumina) were used to isolate poly(A)-containing mRNA molecules. Beads were warmed to room temperature and washed three times with lysis/binding buffer (20 mM Tris-HCl, pH 7.5, 500 mM LiCl, 0.5% LiDS, 1 mM EDTA, 5 mM DTT) and re-suspended in the original starting volume with lysis/binding buffer. We then added 50 μl of washed beads to each well of eluted RNA from above and followed the manufacturer's protocol (Illumina TruSeq Stranded mRNA Sample Preparation Guide). Instead of using Illumina's Fragment, Prime, Finish Mix, we eluted mRNA from the beads with 16.5 μl of Elution solution (20 mM Tris-HCl pH 7.5, 1 mM EDTA). A measure of 1 μl of 1:5000 diluted ERCC Ex-Fold Spike-Ins (ThermoFisher) was added to half of the sample wells. We used this as input for the automated PLATE-Seq protocol, beginning with the addition of 5× Protoscript RT Buffer and 1.5 μl of 100 μM adapter-linked, barcoded oligo(dT) primers to each well.

**PLATE-Seq data processing**. PLATE-Seq data comprise sets of two paired-end reads. The first read contains an eight-base barcode sequence that identifies the well from which a given sample originates. The second read contains a sequence that typically maps strand specifically to the 3′ end of an mRNA transcript. We first record the barcode sequence associated with each read. The barcode sequences are designed so that a single error can be corrected, and so we allow an edit distance of one from the actual barcode sequences. Next, we use the bwa-mem aligner to map the set of second reads to a pre-assembled human transcriptome (hg19, UCSC known genes). We then demultiplex the resulting alignments and count the number of uniquely mapped reads associated with each gene. We define a read as uniquely mapped if the maximum alignment score among alignments that map to the correct strand is associated with a single gene.

**Gene expression analysis from PLATE-Seq**. The differential expression analysis shown in Fig. 2i was conducted using DESeq2 using a negative binomial model[18]. Differentially expressed genes ($q < 0.05$) from all comparisons of drug-treated vs. vehicle control samples were identified exclusively from the PLATE-Seq data. A log-scale fold-change matrix was then generated for this set of differentially expressed genes across all PLATE-Seq and TruSeq samples and clustered using multidimensional scaling (MDS) with Euclidean distance as implemented in MATLAB.

For all other analyses, gene expression signatures for drug perturbations were obtained by first variance stabilizing the raw read counts with DESeq2 (ref. [18]) and then subtracting the mean vehicle control expression level from the perturbation expression levels.

**Regulatory network analysis from PLATE-Seq**. Gene expression signatures were computed for each individual sample by first variance stabilizing the gene expression data with the DESeq package, and then subtracting the average of the vehicle control samples from each drug treatment profile. Relative protein activity was inferred from the single-sample gene expression signatures using the VIPER algorithm[19].

**Large-scale PLATE-Seq screen in U87 cells**. U87 cells were plated in a 384-well plate at 200,000 cells ml⁻¹ in 20 μl of media for a final count of 4000 cells per well. After 24 h, the Labcyte Echo 550 Liquid Handling system was used to dispense drugs at concentrations resulting in 20% growth inhibition (Supplementary Table 1). After 24 h of drug treatment, media was removed and cells were washed with PBS. Cells were then lysed in 20 μl of Buffer TCL (Qiagen) containing 1% β-mercaptoethanol. Following the z-pattern of quadrants for a 384-well plate, 20 μl of lysate from quadrant 1, and 20 μl of lysate from quadrant 3 were pooled into one well of a 96-well TurboCapture plate (mRNA TurboCapture Plate, Qiagen) for a total of 40 μl of lysate in each well during the mRNA capture step. Automated PLATE-Seq was then followed as above. The resulting pooled libraries were sequenced on an Illumina HiSeq 4000 using a 50-cycle SBS kit and a PE cluster kit. Read 1 comprised 18 cycles and Read 2 comprised 50 cycles.

**Comparison of PLATE-Seq screens to CMap and LINCS**. In Fig. 3, we compare the coefficient of variation (CV) between PLATE-Seq, CMap, and LINCS screening data sets. For PLATE-Seq, we took our U87 screen (4 × 96 wells with 184 drugs plus controls) data set and computed the CV and average expression for each gene in each duplicate measurement. We then plotted the average CV vs. average expression across all duplicates for each gene in Fig. 3a–c and distribution of these same average CVs in Fig. 3d–f. For CMap, we took all duplicates from screens conducted in MCF7 (processed average difference values available on the Gene Expression Omnibus under accession GSE5258) and applied the same calculations as we did for PLATE-Seq. We analyzed microarray data generated using the GPL3921 and GPL96 platforms separately. Finally, we obtained the LINCS data set from the Gene Expression Omnibus under accession GSE70138 (Level 2 data corresponding to gene expression values for genes that were measured directly) for

~26,000 randomly selected duplicates and applied the same calculation to the ~1000 genes measured directly on the L1000 platform. In Supplementary Fig. 4, we compare PLATE-Seq, CMap, and LINCS in terms of the fold change in each gene expression measurement across all perturbations relative to their respective vehicle controls. We calculated these distributions using the same expression matrices that we used for the CV comparisons.

**Note on cells lines**. We note that BT20 cells are on the ICLAC list of commonly misidentified cell lines, but that the identity of a particular cell line is immaterial to the conclusions of our technical assessment of PLATE-Seq. Nonetheless, the BT20, U87, and LnCAP cell lines used these studies were all obtained from ATCC, which authenticates the cell lines it provides by STR testing. Internally, we tested all of the cell lines for mycoplasma prior to use.

**Code availability**. The VIPER algorithm is available at http://califano.c2b2.columbia.edu/viper/.

**Data availability**. All of the raw and processed data associated with this study are available at the Gene Expression Omnibus under accession GSE97460.

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

## Acknowledgements

P.A.S. was supported by Grant K01EB016071 from NIH/NIBIB. A.C. was supported by grant R35CA197745 from NIH/NCI. A.C. was also supported by grants S10OD021764 and S10OD012351 from the Office of the Director of NIH, and both A.C. and P.A.S. were supported by grant U54CA209997 from NIH/NCI.

## Author contributions

PLATE-Seq was conceived and designed by A.C. and P.A.S. Experimental methodology for PLATE-Seq was developed by E.C.B., F.R. and P.A.S. Analytical methodology for network-based analysis of PLATE-Seq was developed by M.J.A. and A.C. Data processing and analysis were conducted by E.C.B., F.R., M.J.A., P.A.S. and A.C. HTS experiments were conducted by F.R., R.R., H.L. and C.K. Automated PLATE-Seq was developed and implemented by R.R., C.K. and E.C.B. E.C.B., F.R., M.J.A., R.R., H.L., C.K., A.C. and P.A. S. wrote the paper.

## Additional information

**Competing interests:** M.J.A. is chief scientific officer of DarwinHealth Inc. A.C. is a founder of DarwinHealth Inc. The remaining authors declare no competing financial interests.

