## [Peer Review file · Nature Communications]

Reviewers' comments:

Reviewer #1 (Remarks to the Author):

Bush et al. describe a new methodology they term PLATE-Seq that enables low cost RNA-Seq at scale. The appeal of this approach is for characterization of genetic or pharmacological hits that arise in high throughput screens. A similar logic motivated the L1000 project at the Broad Institute. However, despite the similarity between these approaches the simplicity and systematic approach taken here that does not require preselection of probes is highly appealing. Furthermore, although recent reports used a somewhat similar strategy for high throughput single cell RNA Seq (Dixit et al. Cell 2016 and Adhemar et al. Cell 2016) these require specialized equipment for library preparation and are not suitable for many lower throughput applications. Thus, PLATE-Seq is a highly relevant technique with applications in functional genomics and drug characterization and are of interest to a broad scientific community. However, I believe that some details should be addressed before publication in Nature communications.

Major comments

1. The methodology for PLATE-Seq described by the authors includes three steps that are different from conventional methods used in RNA-Seq. The main differences are; a) mRNA is extracted using TCL buffer and oligo dT plates. b) cDNA preparation without fragmentation and adaptor ligation (barcodes are added during cDNA synthesis). c) Low depth sequencing (1-2M reads/sample as oppose to 30M reads/sample).

To evaluate this new methodology the authors compare PLATE-Seq to conventional RNA-Seq (TrueSeq). However, from this comparison it is hard to assess the different steps of PLATE-Seq. It would be very useful to assess the different steps of PLATE-Seq and to add a comparison of samples that mRNA was prepared using conventional methods (for example Trizol and polyT beads) followed by cDNA preparation and sequencing using the PLATE-Seq protocol. Since methods for mRNA preparation have a great effect on the quality of RNA-Seq it would be important to see if this would change PLATE-Seq quality in a dramatic way.

2. The authors show that replicates in PLATE-Seq are correlated however this is not an impressive correlation ($r=0.2$). I assume that this correlation uses the difference in expression compare to DMSO as a metric (and not raw counts). However, it would be helpful to add a heatmap or plot showing the correlation between replicates (using RPKM). Also, to allow readers a fair comparison the authors should add to the plots in Fig. 3 replicate correlations using True-Seq and 30M reads.

3. In Fig. 3 the authors performed an impressive experiment that includes the use of PLATE-Seq to profile 92 drugs. The potential use of PLATE-Seq would be, similar to the L1000 project, to identify expression profiles that are similar (using PCA or hierarchical clustering for example). However, no such analysis is presented.

Minor comments

1. The sequences of primers used in PLATE-seq should be provided.

2. To facilitate the evaluation of PLATE-Seq by independent laboratories the raw read counts obtained in the experiments described here should be provided.

Reviewer #2 (Remarks to the Author):

The manuscript by Bush et al. describes a novel technology for inference of generalized pathway activity that is cost effective for high throughput screening experiments. Appropriate reference is given to comparable technologies. The method seems well developed in that it comprises both wet

and computational protocols to achieve the goal. Further, appropriate experiments have been performed to demonstrate performance of the technology.

A weakness of the current manuscript are the analytical procedures used to compare technologies that are illustrated in figure 3. The comparison of technologies is somewhat subjective, and the measure chosen (correlation) is not well suited to the task. This section could be strengthened by objective evaluation of repeatability and signal to noise of each platform. I offer a suggestion below, but certainly there are other methods for objective evaluation that would improve this section.

Technical replicates are available, thus one could estimate the coefficient of variation at the gene level (or of the inferred activity) for each platform. Then an objective comparison of repeatability (measured by CV distributions) can be performed. The non-replicates provide a point of contrast. Again, gene level statistics (or inferred differential protein activity) can be assessed for each platform. These differential measures can be compared across platforms and slopes can be estimated to objectively compare signal to noise.

Response to Reviewers

Original reviewer comments appear below in *italics* with responses in **green**.

Reviewer 1

Bush et al. describe a new methodology they term PLATE-Seq that enables low cost RNA-Seq at scale. The appeal of this approach is for characterization of genetic or pharmacological hits that arise in high throughput screens. A similar logic motivated the L1000 project at the Broad Institute. However, despite the similarity between these approaches the simplicity and systematic approach taken here that does not require preselection of probes is highly appealing. Furthermore, although recent reports used a somewhat similar strategy for high throughput single cell RNA Seq (Dixit et al. Cell 2016 and Adhemar et al. Cell 2016) these require specialized equipment for library preparation and are not suitable for many lower throughput applications. Thus, PLATE-Seq is a highly relevant technique with applications in functional genomics and drug characterization and are of interest to a broad scientific community. However, I believe that some details should be addressed before publication in Nature communications.

We thank the reviewer for these positive comments.

Major comments

1. The methodology for PLATE-Seq described by the authors includes three steps that are different from conventional methods used in RNA-Seq. The main differences are; a) mRNA is extracted using TCL buffer and oligo dT plates. b) cDNA preparation without fragmentation and adaptor ligation (barcodes are added during cDNA synthesis). c) Low depth sequencing (1-2M reads/sample as appose to 30M reads/sample).

To evaluate this new methodology the authors compare PLATE-Seq to conventional RNA-Seq (TrueSeq). However, from this comparison it is hard to assess the different steps of PLATE-Seq. It would be very useful to assess the different steps of PLATE-Seq and to add a comparison of samples that mRNA was prepared using conventional methods (for example Trizol and polyT beads) followed by cDNA preparation and sequencing using the PLATE-Seq protocol. Since methods for mRNA preparation have a great effect on the quality of RNA-Seq it would be important to see if this would change PLATE-Seq quality in a dramatic way.

We agree with the reviewer that comparing the performance of the PLATE-Seq library construction procedure with conventionally processed mRNA (e.g. via column isolation and oligo(dT) bead purification) to our plate-based purification method would be very valuable. To carry out the suggested comparison, we chose column- and oligo(dT)-based isolation here as our point of comparison because we applied this method in our original comparison to the Illumina TruSeq system. We processed a 96-well plate in which half of the material in each well was subjected to our normal PLATE-Seq workflow and the other half was processed using column-based total RNA isolation and oligo(dT) bead-based mRNA purification followed by PLATE-Seq library construction. As you can see, in **Supplementary Figure S3,**

the performances as evaluated by gene detection efficiency and saturation behavior are essentially identical. Hence, we conclude that the plate-based mRNA isolation procedure implemented here, which is much less expensive and more amenable to automation than conventional column- and bead-based isolation, does not have a significant effect on data quality.

2. The authors show that replicates in PLATE-Seq are correlated however this is not an impressive correlation ($r=0.2$). I assume that this correlation uses the difference in expression compare to DMSO as a metric (and not raw counts). However, it would be helpful to add a heatmap or plot showing the correlation between replicates (using RPKM). Also, to allow readers a fair comparison the authors should add to the plots in Fig. 3 replicate correlations using True-Seq and 30M reads.

Both Reviewers 1 and 2 felt that our correlation-based analysis was a weakness of the manuscript, and so we have eliminated this analysis. In order to respond to comments from Reviewer 2, we have added a more sophisticated analysis of gene-level noise across replicates. We note that conducting an accurate assessment of Illumina TruSeq on a similar scale is prohibitively expensive (a few hundred thousand dollars) and beyond the scope of this work. For additional details on our improved analysis in **Figure 3**, please see the response to the comments from Reviewer 2 below.

3. In Fig. 3 the authors performed an impressive experiment that includes the use of PLATE-Seq to profile 92 drugs. The potential use of PLATE-Seq would be, similar to the L1000 project, to identify expression profiles that are similar (using PCA or hierarchical clustering for example). However, no such analysis is presented.

We agree with Reviewer 1 regarding the potential value of inferring gene expression programs from this large-scale screen. In fact, in the newest version of the manuscript, we now include data from an even larger screen. However, we feel that deeper analysis of this screen is beyond the scope of the current manuscript, which focuses mainly on a technical assessment of this new method. We intend to conduct the proposed analysis on this and many additional screens that we have carried out recently in a future, forthcoming manuscript.

Minor comments

1. The sequences of primers used in PLATE-seq should be provided.

We completely agree and apologize for this omission. A complete list of primer sequences is now reported in **Supplementary Table S2**.

2. To facilitate the evaluation of PLATE-Seq by independent laboratories the raw read counts obtained in the experiments described here should be provided.

We completely agree and have submitted all of our raw and processed data to the Gene Expression Omnibus. It is now publically available and can be viewed under accession **GSE97460** as noted in the latest version of the manuscript.

Reviewer 2

The manuscript by Bush et al. describes a novel technology for inference of generalized pathway activity that is cost effective for high throughput screening experiments. Appropriate reference is given to comparable technologies. The method seems well developed in that it comprises both wet and computational protocols to achieve the goal. Further, appropriate experiments have been performed to demonstrate performance of the technology.

We thank the reviewer for these positive comments.

A weakness of the current manuscript are the analytical procedures used to compare technologies that are illustrated in figure 3. The comparison of technologies is somewhat subjective, and the measure chosen (correlation) is not well suited to the task. This section could be strengthened by objective evaluation of repeatability and signal to noise of each platform. I offer a suggestion below, but certainly there are other methods for objective evaluation that would improve this section.

Technical replicates are available, thus one could estimate the coefficient of variation at the gene level (or of the inferred activity) for each platform. Then an objective comparison of repeatability (measured by CV distributions) can be performed. The non-replicates provide a point of contrast. Again, gene level statistics (or inferred differential protein activity) can be assessed for each platform. These differential measures can be compared across platforms and slopes can be estimated to objectively compare signal to noise.

We agree with the reviewer that there is ample room for improvement in our original correlation-based assessment and comparison to alternative methods. We also thank the reviewer for providing a concrete suggestion on how to improve our analysis. To address these comments, we have taken two key steps: 1) we now include a larger-scale and improved PLATE-Seq screening data set (this time on U87 cells treated with 184 different drugs) and 2) we computed the gene-level CV distributions for this new PLATE-Seq screen, CMAP data from two different microarray platforms, and LINCS data. We find that PLATE-Seq performs comparably to CMAP in terms of gene-level noise and somewhat worse than LINCS. However, we note that the processed CMAP average-different data that are publically available display some unusual noise characteristics. For one, the relationship between expression level and noise is not monotonic. Furthermore, for one of the two platforms the noise distribution is actually bimodal. Finally, we note that although LINCS outperforms PLATE-Seq somewhat in terms of gene-level noise, LINCS is measuring <1,000 select genes whereas PLATE-Seq is genome-wide, which sets the bar considerably higher. In any case, we feel that this new analysis represents a significant improvement to the manuscript and thank the reviewer for the detailed assessment and suggestion.

Reviewers' comments:**Reviewer #1 (Remarks to the Author):**

The authors have satisfactorily addressed the questions and comments raised in the review. I believe the manuscript is suitable for publication with no further modifications.

Reviewer #2 (Remarks to the Author):

The comparisons of CVs are interesting and permit proper comparison of repeatability across the competing technologies. As noted in the previous response, this should be contrasted with a comparison of signal across the same platforms. More specifically, repeatability is only one part of the equation and while the comparative data looks great, CVs may simply be low due to low signal.

The authors should provide a signal assessment using the same data in the CV estimates. This would take the form of calculating differential signal (i.e. fold change) between representative conditions. Then comparing fold changes of PLATE-seq to fold changes from the comparable technologies. Slope estimates from this comparison will allow accurate determination of compression or increase of signal in PLATE-seq versus other technologies. See figure 6 of the primary MAQC paper (PMID: 16964229) for an example, although slopes instead of correlation measures would be preferred.

Response to Reviewers

Original reviewer comments appear below in *italics* with responses in **green**.

Reviewer 1

The authors have satisfactorily addressed the questions and comments raised in the review. I believe the manuscript is suitable for publication with no further modifications.

We thank the reviewer for these positive comments.

Reviewer 2

The comparisons of CVs are interesting and permit proper comparison of repeatability across the competing technologies. As noted in the previous response, this should be contrasted with a comparison of signal across the same platforms. More specifically, repeatability is only one part of the equation and while the comparative data looks great, CVs may simply be low due to low signal.

The authors should provide a signal assessment using the same data in the CV estimates. This would take the form of calculating differential signal (i.e. fold change) between representative conditions. Then comparing fold changes of PLATE-seq to fold changes from the comparable technologies. Slope estimates from this comparison will allow accurate determination of compression or increase of signal in PLATE-seq versus other technologies. See figure 6 of the primary MAQC paper (PMID: 16964229) for an example, although slopes instead of correlation measures would be preferred.

We cannot conduct the analysis in Figure 6 of the MAQC paper exactly as suggested by the reviewer because that figure was generated using data in which two different platforms were used to measure gene expression changes (microarray and qPCR) across the same perturbations in the same biological context. This analysis requires a direct, gene-level comparison between perturbations using our PLATE-Seq technology and perturbations made in the original CMap and LINCS publications. Those studies involved different cell lines, drugs, and experimental conditions from what we employed in our PLATE-Seq screens, and so a gene-level comparison or correlation between effect sizes will not be meaningful. However, to address the reviewer's concerns, we have performed a virtually equivalent analysis. We now provide the distributions of fold-changes for each gene expression measurement relative to the appropriate vehicle control across all perturbations in our PLATE-Seq screen, the two CMap data sets, and the LINCS data set in Supplementary Figure S4.

REVIEWERS' COMMENTS:

Reviewer #2 (Remarks to the Author):

Thank you for the clarification regarding comparisons of differential signal. The authors have responded to all my concerns.